# In Situ Fabrication of High Dielectric Constant Composite Films with Good Mechanical and Thermal Properties by Controlled Reduction

**DOI:** 10.3390/molecules28062535

**Published:** 2023-03-10

**Authors:** Zhaoyu Hu, Lian Chen, Yongmei Zhu, Chunmei Zhang, Shaohua Jiang, Haoqing Hou

**Affiliations:** 1Department of Chemistry and Chemical Engineering, Jiangxi Normal University, Nanchang 330022, China; 15270586286@163.com (Z.H.); zhuym0506@jxnu.edu.cn (Y.Z.); 2Jiangsu Co-Innovation Center of Efficient Processing and Utilization of Forest Resources, International Innovation Center for Forest Chemicals and Materials, College of Materials Science and Engineering, Nanjing Forestry University, Nanjing 210037, China; lianchen@njfu.edu.cn; 3Institute of Materials Science and Devices, School of Materials Science and Engineering, Suzhou University of Science and Technology, Suzhou 215009, China; cmzhang@usts.edu.cn

**Keywords:** polyimide, reduce graphene oxide, composites, dielectric property, thermal property, mechanical property

## Abstract

As a common two-dimensional carbon material, graphene has been widely doped into polymers to prepare high-performance dielectric materials. However, the shortcomings of graphene, such as large specific surface area and poor dispersion, limit its further application. Therefore, in this work, to solve the problem regarding the uniform dispersion of graphene in the matrix, in situ polymerization was used to prepare graphene/polyimide films, in which 1,4-diiodobutane was used as a reduction agent to prevent the aggregation of graphene oxide (GO) during imidization. High dielectric constant composite films were obtained by adjusting the ratio of 1,4-diiodobutane in GO. The results show that the resulting graphene/polyimide composite film possessed a dielectric constant of up to 197.5, which was more than 58 times higher than that of the polyimide (PI) film. Furthermore, compared to the pure PI film, the composite films showed better thermal stability and mechanical properties. Thermal performance tests showed that the 1,4-diiodobutane added during the preparation of the composite film was thermally decomposed, and there was no residue. We believe our preparation method can be extended to other high dielectric composite films, which will facilitate their further development and application in high power density energy storage materials.

## 1. Introduction

As the core material of electronic devices, high-performance dielectric materials have become a critical factor in developing electronic devices. Therefore, dielectric materials with high dielectric constants, good mechanical properties, high temperature resistance, high energy storage, and other advantages have become the focus of current research [1,2,3,4,5]. Although traditional dielectric ceramics have a high dielectric constant and excellent thermal stability, their shortcomings, including high brittleness, high processing temperature, and low dielectric breakdown strength, limit their applications [6,7,8,9]. On the contrary, polymers with lower dielectric constants are known for their excellent mechanical flexibility and ease of processing [10,11,12,13]. Graphene is an excellent filler for preparing high dielectric materials because of its excellent electrical conductivity, high temperature resistance, high strength, and chemical stability [14,15,16]. However, its large specific surface area, poor dispersion performance, and easy agglomeration limit its application range [17,18,19,20,21,22]. Therefore, solving the problem of evenly dispersing graphene in the matrix has become a research hotspot in recent years. Stable dispersion systems can be formed by covalent or non-covalent interactions with polymers. There are no functional groups in the graphene structure, so it is difficult for it to form strong interactions with polymers directly, while as a derivative of graphene [23,24,25], the surface of GO includes oxygen-containing functional groups such as carboxyl, hydroxyl and epoxide, which leads to good dispersion in the polymer matrix and offers the possibility of preparing graphene/polymer composite films. However, these groups can disrupt the conjugated conductive structure of grapheme [26], which needs to be reduced to reduced graphene oxide (rGO) after homogeneous dispersion in order to obtain high dielectric constants [27,28].

Ramachandran used NaBH_4_ as the reducing agent to reduce GO powder at 90 °C for 1 h [29]. The GO in suspension was effectively reduced by NaBH_4_, and the GO was layered with uniform structure because of agglomeration. However, NaBH_4_ could only reduce C=O to C-OH, which had a reduced performance for hydrogenation, the oxygen element was not eliminated, and no new C=C was generated. The reduction of C=O did not directly restore the conjugated structure of graphene. Ruoff’s team took the lead in proposing the preparation of graphene by reducing GO [30]. In this report, the carbon/oxygen element mass ratio (C/O ratio) was used as an indicator to confirm the reduction degree. The results showed that the C/O ratio had been increased from 2.7 to 10.3, and the conductivity reached 2420 S/m. However, hydrazine hydrate was highly toxic and costly, and graphene was prone to agglomerate after reduction, which seriously limits its application. In 2010, Ruoff’s team published the first report on the reduction of GO by HI [31]. The whole reduction process was carried out in an HI solution, and its reduction ability was not weaker than that of N_2_H_4_. After reduction, the C/O ratio reached 11.5, but a small amount of I^−^ and I^2^ were present in the graphene structure after reduction [32]. In 2015, Haiquan Guo et al. used 1,2-diiodoethane to prepare graphene (IGO)/polyimide (PI) composites. The composite films exhibited a conductivity of 2.22 S/m in this way; however the reducing agent needed to be dissolved with a polar solvent and then heated at 80 °C for 12 h for the reduction to take effect.

Herein, in this work, to achieve the better reduction of GO and prevent the agglomeration of reduced products, 1,4-diiodobutane and GO were used as the reducing agent and conductive filler, respectively, to prepare high dielectric composite films [33]. PI with high temperature resistance, chemical stability, and excellent mechanical properties was used as the matrix polymer [34,35,36]. Notably, the use of 1,4-diiodobutane as a reducing agent had three advantages: (1) a strong reducing ability, without destroying the macrostructure of graphene after reduction; (2) a low working temperature, allowing for ease of operation; (3) the C-I single bond had low bond energy and is easy to break, which was beneficial to reduce the residual amount of heteroatoms. In this paper, reduced graphene oxide/polyimide (rGO/PI) composite films were prepared by a simple in situ polymerization method. The degree of GO reduction was successfully adjusted by varying the amount of 1,4-diiodobutane added. The high dielectric constant, low dielectric loss, high strength, and high temperature resistance of the composite films were investigated by analyzing the variation of the dielectric properties, mechanical properties, and heat resistance. In addition, the effect of the degree of GO reduction on the properties of the rGO/PI composite films was discussed. We believe that these high dielectric constant composite films with good mechanical and thermal properties can provide new insights into the development of capacitors and filters, along with the miniaturization and integration of electronic devices.

## 2. Results and Discussion

### 2.1. Characterization of GO and rGO

The number of functional groups at the base and edge of rGO varies with the different weight ratios of GO and 1,4-diiodobutane. The reduction degree of GO was measured using laser confocal micro-Raman spectroscopy, ultraviolet-visible, and XPS. Raman spectroscopy can quickly and effectively characterize the structure of carbon materials, and can infer ordered and disordered carbon crystal structures. Figure 1a shows the Raman spectra of GO and rGO treated with different proportions of reducing agents. It can be seen from the figure that rGO has two strong absorption peaks at 1320 cm^−1^ and 1580 cm^−1^, respectively. Peak D is the acoustic vibration peak induced by disordered carbon. The intensity of the peak can be used to indicate the disorder and structural imperfection of carbon materials. The G peak is caused by the first-order Raman scattering of the E_2g_ phonon of the sp^2^ carbon atom, and the intensity of the peak can be used to indicate the integrity and order of the carbon material. Peak D and Peak G, respectively, represent the number of sp^3^ and sp^2^ hybrid carbon atoms, and the intensity ratio (I_D_/I_G_) can be used to indicate the disorder degree of the carbon material [37]. After treatment with different proportions of the reducing agent (1,4-diiodobutane: GO = 0, 20, 40, 60, 80), the intensity ratios of rGO are 0.86, 1.18, 1.50, 2.13, 3.76, respectively; that is, as the amount of reducing agent increases, the I_D_/I_G_ ratio of the obtained graphene gradually increases. Theoretically, when the GO is reduced, the functional groups on the graphite flakes will be removed, the order of the sp^2^ carbon network structure will increase, the sp^2^ area will become larger, and I_D_/I_G_ will decrease. The possible reason for this change trend, which is contrary to theoretical predictions, is that after GO is reduced, a large number of sp^3^ hybrid carbon atoms will re-form new sp^2^ hybrid regions after deoxidation, and the re-formed sp^2^ regions are greater than the GO [38,39].

The UV-Vis spectra of GO suspension show a characteristic peak at 245 nm, which corresponds to the π-π * transition of the aromatic C=C bond in GO [40]. However, as the proportion of the reducing agent increases, the absorption peaks belonging to the π-π * transition gradually redshift to higher wavelengths, redshifting to 252 nm, 256 nm, 260 nm, and 268 nm, respectively (Figure 1b). This shows that during the chemical reduction process, the density of π-conjugated electrons continues to increase, and the conjugated electronic structure of graphene is gradually restored [41,42]. This means that the reduction degree of rGO can be customized by adjusting the ratio of GO and the reducing agent.

As the proportion of reducing agents increased, the peak intensity of C 1s increased significantly. Moreover, the ratio of C to O increased from 2.21 to 5.72 (Figure 1c). These results suggested that 1,4-diiodobutane was able to effectively control the conversion of GO to rGO under mild reaction conditions, without using strong acids. Figure 1d–f showed the deconvolution spectra of GO, rGO-20, and rGO-80. Each spectrum of C 1s showed three prominent peaks, which correspond to three groups: graphitic carbon sp^2^, epoxy/hydroxyl, and carboxylate. Upon increasing the dosage of the reducing agent, there was a gradual decline in peak intensities for the C-O and O-C=O groups, indicating that each group was gradually removed with a degree of reduction. Hence, it could be concluded that the degree of reduction of rGO could be genuinely controlled via increasing or decreasing the amount of reducing agent. This conclusion is consistent with the Raman, UV-Vis, and IR spectra results. Compared to the unreduced GO, all rGOs had a relatively high C/O ratio, suggesting that they were partially reduced and well dispersed in the polymer matrix. By in situ chemical reduction, 1,4-diiodobutane could be an ideal reducing agent for preparing graphene-based polymer composite films, especially those made from hydrazine-sensitive materials (e.g., PI). The reduction mechanism might be due to the formation of hydrogen iodide in the reaction process of eliminating 1,4-diiodobutane by heating. The iodide anion was the primary reducing agent, and it is integral to the reduction process, which can provide electrons to facilitate the recovery of the conjugated structure of grapheme [43].

### 2.2. Characterization of rGO/PI Composite Films

According to prior experience, when GO nanosheets are loaded with 1.5 wt.%, the conductive network is formed by in situ reduction, which is considered as the threshold of permeation. Therefore, it is feasible to prepare rGO/PI composite films by controlling the reduction process and selecting a relatively low filler loading of 1.0 wt.% of GO. FTIR spectra were used to analyze the chemical structure of the rGO and the rGO/PI composite films. Figure 2a shows the infrared spectra of GO and rGO. The FTIR spectra of GO show typical absorption peaks at 3410, 1716, 1580, and 1217 cm^−1^, corresponding to contraction vibrations in hydroxyls, C=O stretching in the -COOH, and C=C stretching in unoxidized C-C bones, as well as the stretching vibrations in epoxy groups on the GO sheets [44]. The degree of reduction of graphene increases with the proportion of 1,4-diiodobutane in the reduction process, and the characteristic peaks of the functional groups progressively weaken [45]. In addition, as the degree of reduction increases, the hydroxyl peak shows a blueshift trend. Due to the small percentage of water present in the test sample, intermolecular hydrogen bonds are formed between water molecules and graphene and between the graphene sheets, along with intramolecular hydrogen bonds between the graphene sheets. The increase in the degree of reduction leads to a decrease in the hydroxyl and carboxyl groups of graphene, which leads to a decrease in intramolecular and intermolecular hydrogen bonds, which broadens the infrared band of the hydroxyl peak and causes a blueshift. Therefore, the GO can be reduced in a controlled way [46]. Figure 2b showed the FTIR spectra of the PI film and the rGO/PI composite films, respectively. PI showed two characteristic absorption peaks at 1773 cm^−1^ and 1711 cm^−1^, and the same peaks appeared in the rGO/PI composite films, while the other peaks were not significant [47]. The wavenumbers, intensities, and corresponding groups of all absorption bands in the infrared spectra are shown in Table 1. The rGO/PI films were obtained by treatment with 1.0 wt.% of the GO reducing agent; after heat treatment at 420 °C, GO was further reduced and the reducing agent component was utterly decomposed.

Figure 3a showed the TEM image of GO. There were wrinkles around the edges of the GO layers, revealing that there were extremely few GO layers. In addition, the TEM image showed that the surface of GO was flat, smooth, and transparent. A TEM image of rGO is depicted in Figure 3b. It can be seen from the figure that there are parts of the graphene sheets that overlap, in undulating folds like waves, with a semi-transparent structure that resembled yarn. This was because the oxygen-containing graphene functional groups were reduced, resulting in intermolecular forces between the graphene layers. However, it is quite easy to reunite the slices, and the overlap between them is readily apparent. The cross-sections of the PI film and the composite films were examined by scanning electron microscopy in order to better understand the interfacial interactions between rGO with varying reduction extents and the PI matrix and to evaluate their dispersion and compatibility in a polymer matrix. The fracture surface of pure PI was flat and smooth compared to the surrounding matrix, indicating characteristics typical of brittle fractures (Figure 3c). In Figure 3d–f, it can be seen that compared with the PI film, the fracture surfaces of the composite films were relatively rough, which indicated that GO and PI had good interfacial interaction and GO was well dispersed in the matrix. Compared with GO/PI, the cavities of rGO/PI composite films tend to be more prominent, resulting from the pulling out of rGO.

### 2.3. Dielectric Properties of rGO/PI Composite Films

The dielectric performances of the composite films are shown in Figure 4a–d. The dielectric constant increased as the weight ratio (GO/1,4-diiodobutane, G/D) rises from 1:5 to 1:80. The dielectric permittivity was 197.5 at a frequency of 100 Hz when the G/D was 1:80, approximately 58 times that of PI film. However, the dielectric loss of composite films also increased when GO was reduced in the polymer matrix. Despite this, dielectric loss in composite films of rGO/PI remained remarkably low (0.31). In the preparation of composite films, the reduction of GO further increased with the increase in the 1,4-diiodobutane ratio, partially restoring the structure of the graphene. In fact, a significant amount of the graphene structure and electrical conductivity were restored during the formation of the composite films when GO was reduced. It was reported that the dielectric constant increased, primarily as the result of micro capacitor formation and polarization effects at the interface (also known as the MWS effect) [48]. Due to the layered structure of GO, it could quickly form micro capacitors. Meanwhile, the conductivity of the rGO nanosheets led to more charge carrier accumulation and more MWS polarization; in turn, this led to a high dielectric permittivity [49]. Thus, using GO to prepare composite films with various dielectric constants for a range of dielectric properties can meet practical requirements.

### 2.4. Mechanical and Thermal Properties of rGO/PI Composite Films

As a high temperature resistant material, the thermal stability of PI matrix composite films is essential. The thermal decomposition curves for PI, rGO/PI composite films are shown in Figure 5a, which shows that the thermal weight loss temperatures (T_d_, 5%) of the GO/PI, rGO/PI composite films were higher than those of the pure PI films. As compared to pure PI, the composite films were more stable, which may be because the functional groups of the GO nanosheet layers containing oxygen played a beneficial role as free radical trapping agents, inhibiting the thermal degradation. Additionally, after the reduction of 1,4-diiodobutane, all the composite films showed good thermal stability. After heat treatment at a high temperature (420 °C), the 1,4-diiodobutane added during the preparation of the composite films was thermally decomposed, it did not remain in the composite films, and the thermal stability was not compromised.

In DMTA analysis, the loss tangent curves (tan θ) for the PI and rGO/PI films are shown in Figure 5b for the entire measurement range (50–500 °C), where the tan peak was indicative of the T_g_ for the rGO/PI composite films. The glass transition temperature of the GO/PI composite films is much higher than that of pure PI, which is mainly due to the interaction between the reactive groups of GO and the polymer matrix, resulting in the inability of the polymer chains to move freely, thus increasing the T_g_. The T_g_ of rGO/PI composite films increased from 286.6 °C to 329.2 °C with the increase in the reducing agent dosage. Although the reduction of the reactive groups reduced the interaction with the polymer matrix, the restored graphene structure enhanced the π-π* interaction with the PI molecules due to the intercalation of rGO with the PI matrix. Therefore, the constraining effect of the rGO nanosheet layer on the polymer molecular chain segments also improves the T_g_ of the composite films. The T_5%_ and T_g_ of the rGO/PI composite films are shown in Figure 5c.

Furthermore, the tensile tests for the rGO/PI composite films also reflect the effect of the GO reduction degree on the mechanical properties (Figure 5d). As can be seen from the figure, the tensile strength of pure PI is 181.8 MPa, while the tensile strength of rGO-20/PI is up to 262.0 MPa, which is 1.44 times that of pure PI; the Young’s modulus of rGO/PI is further improved when compared to that of GO/PI, as shown in Figure 5e. This is mainly because of the abundance of reactive groups on GO, which can better disperse in the matrix to produce a strong interface interaction, thus improving the mechanical property of the composite films [50]. Furthermore, the reduction degree of GO would also affect the tensile strength of the whole matrix. However, when GO was further reduced to improve the dielectric properties of the composite films, the interfacial interaction between rGO and the polymer matrix nanosheets was weakened, and the reduction of oxygen functional groups in rGO led to the decrease in the mechanical properties of the composite films. Therefore, the proper degree of reduction of rGO was expected to more effectively improve the mechanical properties of the composite films. Using different particles—1.0 wt.% STRG [12], 1.0 wt.% CFGO [51], 1.0 wt.% RGO-8 [33], 1.0 wt.% rFG [49], and 1.0 wt.% rGO (this work)—as fillers, the dielectric permittivity, mechanical properties, and thermal properties of the PI composite films were compared, as shown in Figure 5f. From the figure, it can be seen that all the properties of the polyimide composite films prepared in this paper are much higher than those previously reported.

## 3. Materials and Methods

### 3.1. Materials

p-Phenylenediamine (p-PDA, 99%) was purchased from Sigma-Aldrich, St. Louis, MO, USA; tetracarboxylic dianhydride (s-BPDA, 99%) was obtained from Aladdin, Baytown, MI, USA, and they were purified by sublimation before use. GO (dispersion in DMF, 10 mg/g) was obtained from Hangzhou Gaoxi Technology, Hangzhou, China; 1,4-diiodobutane (98%) was purchased from Sigma-Aldrich, St. Louis, MO, USA. DMF (99%) was purchased from Tianjin Zhiyuan Reagent, Tianjin, China, purified by CaH_2_ and vacuum distillation, and stored in a 4 A molecular sieve. Other materials were obtained from Aladdin, Baytown, MI, USA, and used as received.

### 3.2. Synthesizing rGO with Variable Degrees of Reduction

First, GO was dispersed into DMF and ultrasound was performed for 24 h. Then, 1,4-diiodobutane was added to the above GO dispersion (0.1 wt.%). After filtering and washing, the rGO was dried at 60 °C for 12 h. In order to prepare rGOs with different reduction extents, the ratio of 1,4-diiodobutane in GO was adjusted from 20 to 40, 60, and 80 and labeled as: rGO-20, rGO-40, rGO-60, and rGO-80.

### 3.3. Preparation of rGO/PI Composite Films

Polymerization in situ was used to prepare all composite films in this study (Figure 6). The following is an example of the preparation process for the rGO/PI composite film: GO dispersion (1.0 wt.%, 6.00 g), p-PDA (1.62 g, 15.0 mmol) and 18 mL DMF were added to a three-necked flask and stirred until completely dissolved. Then, s-BPDA (4.41 g, 15.0 mmol) was added and stirred continuously for 10 h to form a uniform GO/PAA (polyamide acid) solution. Subsequently, 1,4-diiodobutane (0.30 g) was added to the GO/PAA solution and stirred for 2 h. After the above solution was completely dissolved, it was applied to the smooth surface of the glass and heated and imidized, according to the following gradient: 100 °C, 4 h; 150 °C, 2 h; 200 °C, 1 h; 300 °C, 1 h; 350 °C, 1 h; and 420 °C, 1 h. After imidization, the film (rGO-5/PI) was immersed in hot water at 85 °C for 2 h to peel it from the glass substrate. The ratio of 1,4-diiodobutane in GO was adjusted from 5 to 10, 20, 40, 60, and 80 in order to achieve composite films of graphene/polyimide with varying degrees of reduction. The corresponding rGO/PI composites were denoted as rGO-5/PI, rGO-10/PI, rGO-20/PI, rGO-40, rGO-60/PI, and rGO-80/PI, respectively. The above steps could also be used to prepare a GO/PI film and a PI film, without using a reducing agent.

### 3.4. Characterization

A Bruker Tensor 27 FTIR spectrometer was used for the FTIR measurements. The samples were tested in infrared transmission mode, 4000–400 cm^−1^. A small amount of the sample and a sufficient amount of spectral grade potassium bromide powder were obtained, ground thoroughly in an agate mortar, and then slowly poured into an infrared compression mold and pressed to obtain a uniform and transparent sheet. A total of 32 scans were performed at a resolution of 4 cm^−1^, and the spectra obtained were further rubberband baseline corrected, and no normalization was used. The laser Raman of GO and rGO were obtained using a Raman spectrometer with confocal lasers (LabRAM-HR-800), a laser wavelength of 514 nm, and a wavelength scanning range 2000–1000 cm^−1^. The XPS analysis was conducted with an electron spectrometer, ESCALab 250, electron measurement device, Thermo Scientific Corporation, Waltham, MA, USA, and monochromatic radiation, Al Kα, was used. A Lambda 950 UV–Visible spectrophotometer performed ultraviolet-visible spectroscopy in the transmission mode in the 200–800 nm range. A TESCAN vega3 SEM was used to image the fracture morphology of the film at an accelerating voltage of 5 kV. Before observation, gold nanoparticles were spray-coated onto the sample. An electron transmission microscope (JEOL, Tokyo, Japan, JEM-2100) was used to analyze the TEM image. The dielectric properties were measured with the LCR digital bridge (China Tonghui Electronics, Changzhou, China) at room temperature. All samples were glued with conductive silver as electrodes on both sides to avoid unnecessary capacitance and resistance. In the dielectric test, the sample size was 7 mm × 7 mm, while the thickness was 18 μm.

A simple flat plate capacitor is formed using two metal plates that are parallel to each other and separated by a dielectric. A sample is placed between the two plates to measure the capacitance. The relative permittivity of the sample is calculated by the Equation (1):(1)$$C=\varepsilon_{r}S/4\pi kd$$
where *C* is the measured capacitance, is the relative dielectric constant of the sample, *S* is the area of the two parallel electrode plates, *d* is the thickness of the sample, and *k* is the physical constant with a value of 9.0 × 10^9^ N·m^2^/C^2^. The film’s mechanical properties were studied at room temperature using the cmt 0202 electronic universal tester with a speed of 5 mm/mim. Thermal weight loss was measured using a TGA (Perkin Elmer Pyris1), with N_2_ atmosphere, a flow rate 30 mL/min, at 40–800 °C, 10 °C/min, and Dynamic Mechanical Thermal Analyses (DMTA, Rheometric Scientific Inc., Piscataway, MA, USA), with N_2_ atmosphere, a flow rate 30 mL/min, and a test temperature range of 50–450 °C, 3 °C/min, 1 Hz.

## 4. Conclusions

In summary, the rGO/PI composite films were fabricated by in situ polymerization with 1,4-diiodobutane. The composite film with a high dielectric constant was obtained by the controlled reduction of GO. When the weight ratio (G/D) is 1:80, the dielectric constant of the rGO/PI composite film is as high as 197.5, which is more than 58 times higher than that of the pure PI film, and the dielectric loss is only 0.31 at 1000 Hz. The rGO/PI composite film also exhibits good thermal stability. The T_5%_ temperature of the composite film is increased from 557.3 °C (pure PI film) to 596.7 °C, and the T_g_ is increased from 286.6 °C to 329.2 °C. In addition, the tensile strength of the rGO/PI composite film is up to 262 MPa, which is 1.44 times higher than that of the pure PI film. In summary, such an in situ reduction method using 1,4-diiodobutane is designed to provide a new concept for the preparation of high dielectric constant rGO/PI composite films, which can also be extended to other high dielectric composite films, facilitating their further development and application in high power density energy storage materials.

## Figures and Tables

**Figure 1 molecules-28-02535-f001:**
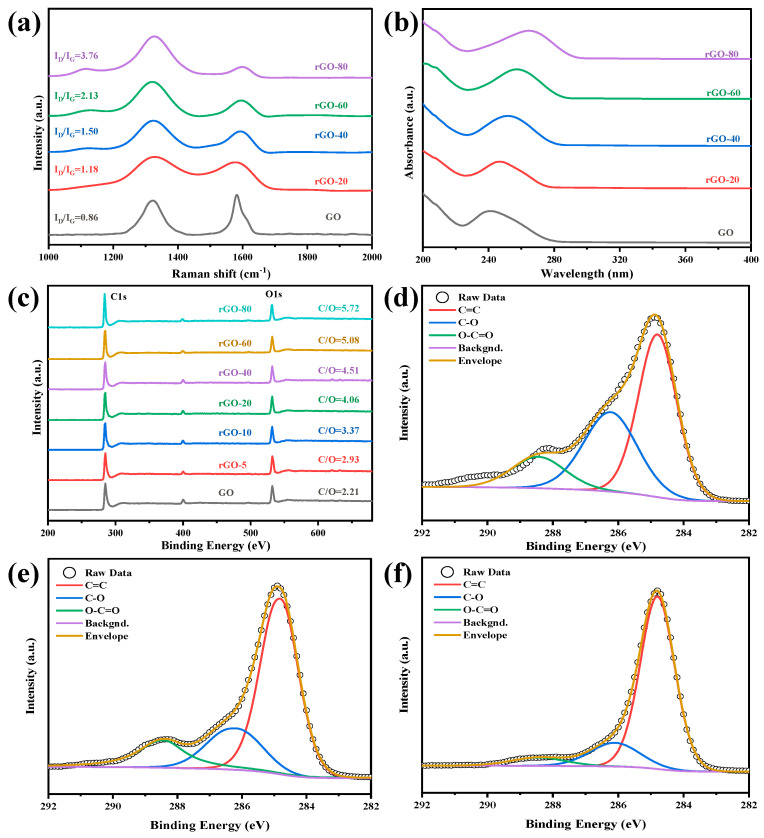
(**a**) Raman spectra of GO and rGO, (**b**) UV-Vis spectra of GO and rGO, with different reduction extent, (**c**) XPS spectra of GO and rGO; C 1s spectra of (**d**) GO, (**e**) rGO-20, and (**f**) rGO-80, in deconvoluted form.

**Figure 2 molecules-28-02535-f002:**
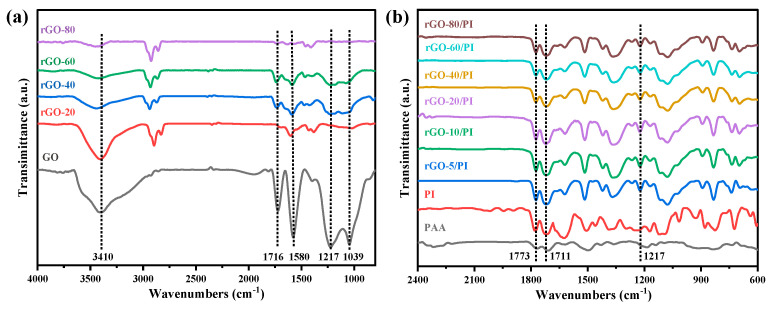
(**a**) FTIR spectra of GO and rGO with different reduction extent, (**b**) FTIR spectra of GO/PI and rGO/PI composite films.

**Figure 3 molecules-28-02535-f003:**
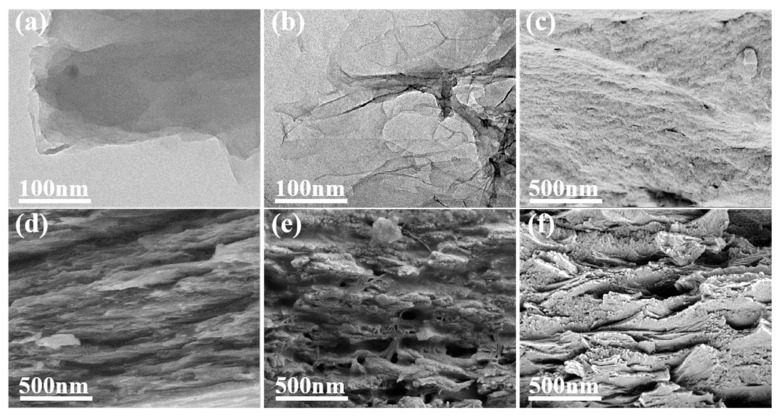
TEM images of (**a**) GO, (**b**) rGO-4; SEM images of the fracture surfaces of (**c**) pure PI, (**d**) GO/PI, (**e**) rGO-40/PI, and (**f**) rGO-80/PI.

**Figure 4 molecules-28-02535-f004:**
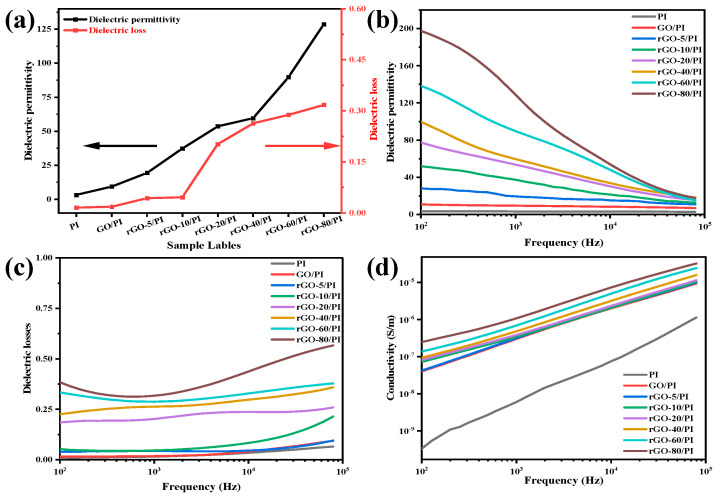
(**a**)Trend diagrams of dielectric permittivity and dielectric loss for pure PI, GO/PI, and rGO/PI composite films at 1000 Hz: frequency dependency, (**b**) dielectric permittivity, (**c**) dielectric loss, (**d**) conductivity.

**Figure 5 molecules-28-02535-f005:**
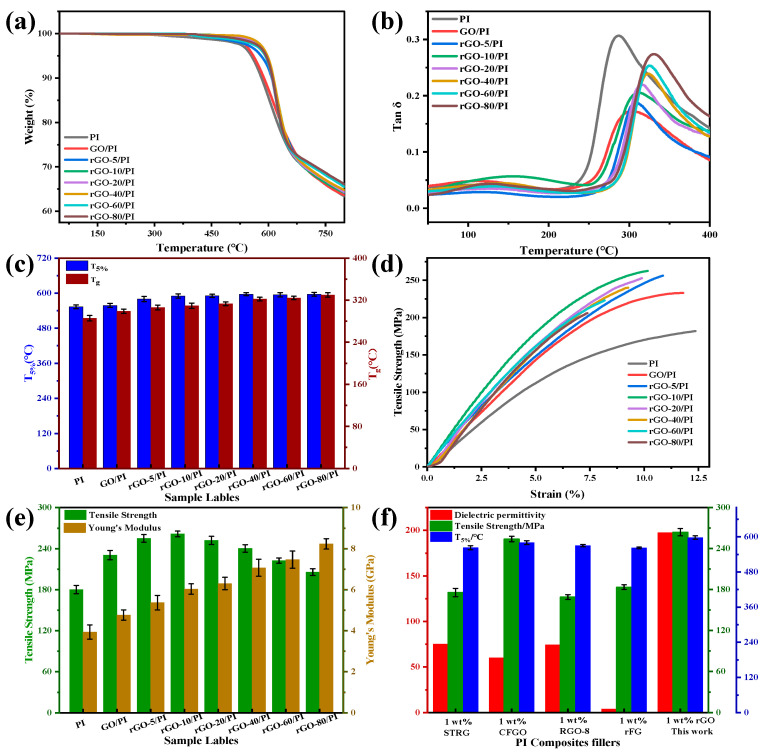
(**a**) TGA curves, (**b**) variation in the loss tangent of the dynamic mechanical properties with temperature, (**c**) the T_5%_ and T_g_, (**d**) typical stress–strain curves, (**e**) the tensile strength and the Young’s modulus for pure PI, GO/PI and rGO/PI composite films, (**f**) the dielectric permittivity, mechanical properties, and thermal properties of various fillers for PI composite films.

**Figure 6 molecules-28-02535-f006:**
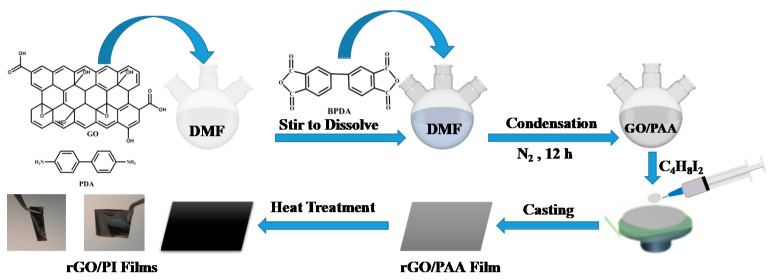
Preparation of rGO/PI composite films.

**Table 1 molecules-28-02535-t001:** Wavenumbers, intensities, and corresponding groups of all absorption bands in the infrared spectra.

Wavenumbers (cm^−1^)	Intensities (%)	Corresponding Groups
3410 ^1^	37	-OH
1773 ^2^	22	N-C=O
1716 ^1^	36	C=O
1711 ^2^	24	-COOH
1580 ^1^	42	C=C
1217 ^1^	48	C-O-C
1039 ^1^	47	C-O

^1^ Marked with the peak of GO. ^2^ Marked with the peak of PI.

## Data Availability

Not applicable.

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
