# Peer review of "In Situ Fabrication of High Dielectric Constant Composite Films with Good Mechanical and Thermal Properties by Controlled Reduction"

_molecules, 2023, doi:10.3390/molecules28062535_

Round 1

Reviewer 1 Report

Dear authors:

In my viewpoint the manuscript ID molecules-2265688  titled "In-Situ Fabrication of High Dielectric Constant Composite Films with  Good Mechanical and Thermal Properties by   Controlled Reduction" can be accept to publication after minor revision.

As suggestion of reviewing:

It is important to cite at materials and Methods item some kind of error or insert some type of standard deviation, or add error bar, at plots.

With relation to infrared spectrum/spectra analysis it is important provide a list (Table) wavenumber and intensity of all absorption bands. Seems that an easy way is add one or more Supplementary file(s).

In the line 313 or near, there is mention to thickness parameter; at moment is reported 18 mm (1.8 cm). I believe that make sense 1.8 mm is more realistic.

Author Response

Response to the general comments: We are very grateful for your positive comments and valuable suggestions. In this manuscript we have prepared graphene/polyimide composite films by in situ polymerization using 1,4-diiodobutane as reducing agent. We have performed the necessary modifications based on your suggestions. We have rewritten the manuscript based on your comments and have marked the revised sections in blue color and we hope that our revised manuscript will be accepted by Molecules.

Comment 1: It is important to cite at materials and Methods item some kind of error or insert some type of standard deviation, or add error bar, at plots.

Reply 1: Thank you very much for your valuable and helpful questions. According to your suggestion, we have added error bar in the graph in the revised manuscript.

Please refer to line 253 on page 7 of the revised manuscript:

“… T5% and Tg of the rGO/PI composite films were shown in Figure 5c. Furthermore, the tensile tests of rGO/PI composite films also reflect the effect of GO reduction degree on mechanical properties (Figure 5d). As can be seen from the figure, the tensile strength of pure PI is 181.8 MPa, while the tensile strength of rGO-20/PI is up to 262.0 MPa, which is 1.44 times that of pure PI, the Young's modulus of rGO/PI are further improved than that of GO/PI, as shown in Figure 5e…Using different particles: 1.0 wt.% STRG, 1.0 wt.% CFGO, 1.0 wt.% RGO-8, 1.0 wt.% rFG, 1.0 wt.% rGO (this work) as fillers, the dielectric permittivity, mechanical properties and thermal properties of PI composite films were significantly compared, as shown in Figure 5f...”

Figure 5. (a) TGA curves, (b) Variation in dynamic mechanical properties loss tangent with temperature, (c) The T5% and Tg (d) Typical stress-strain curves, (e) The tensile strength and the Young's modulus for pure PI, GO/PI and rGO/PI composite films, (f) The dielectric permittivity, mechanical properties and thermal properties of various fillers for PI composite films.

Comment 2: With relation to infrared spectrum/spectra analysis it is important provide a list (Table) wavenumber and intensity of all absorption bands. Seems that an easy way is add one or more Supplementary file(s).

Reply 2: Thank you very much for your valuable and helpful questions. We have provided a table of wavenumber and intensity of all absorption bands in the revised manuscript.

Please refer to line 173 on page 5 of the revised manuscript:

“… PI showed two characteristic absorption peaks at 1773 cm−1 and 1711 cm−1 and the same peaks appeared in the rGO/PI composite films, while the other peaks were not significant. Wavenumbers, intensities and corresponding groups of all absorption bands in the infrared spectra were shown in Table 1…”

Table 1. Wavenumbers, intensities and corresponding groups of all absorption bands in the infrared spectra.

Wavenumbers (cm−1)

Intensities (%)

Corresponding groups

3410 1

37

-OH

1773 2

22

N-C=O

1716 1

36

C=O

1711 2

24

-COOH

1580 1

42

C=C

1217 1

48

C-O-C

1039 1

47

C-O

1 Marked with the peak of GO.

2 Marked with the peak of PI.

Comment 3: In the line 313 or near, there is mention to thickness parameter; at moment is reported 18 mm (1.8 cm). I believe that make sense 1.8 mm is more realistic.

Reply 3: Thank you for your valuable advice. We are sorry for the confusion caused by our writing errors. The thickness of the film is indeed 18 μm, and we have modified it accordingly.

Please refer to line 336 on page 10 of the revised manuscript.

“… In the dielectric test, the sample size was 7 mm * 7 mm, while the thickness was 18 μm…”

Reviewer 2 Report

The presented work is related to the preparation of graphene/polyimide composite films by in situ polymerization using 1,4-diiodobutane as the reducing agent. The article is well written with some minor grammatical errors but I have some questions/comments.

Comment 1: The introduction is well written presenting a good state-of-the-art related to the topic of the work and the references are suitable. However, in the second paragraph in page 2, the authors refer different reducing agents but not 1,2-diiodoethane from reference 47. Is there a reason as why it was not referred? What are the advantages of 1,4-diiodobutane over 1,2-diiodoethane?

Comment 2: The last two sentences of the introduction (line 80 to 83) are conclusions. It would be better to rephrase this part to address the goal of the work.

Comment 3: Regarding the UV-Vis spectra, the shifting of the peak at 245 nm to higher wavelength is clearly evident. However, it is difficult to observe the absorbance increase of the peak as stated. How did the authors detected this occurrence since it does not seem clear from the figure?

Comment 4: Page 3, line 110 Reference to Figure 1b is missing.

Comment 5: In page 3, why could 1,4-diiodobutane be an ideal reducing agent? Reference 47 describes the preparation of graphene (IGO)/polyimide (PI) composites using 1,2-diiodoethane as the reducing agent . How does 1,4-diiodobutane compare to 1,2-diiodoethane? Why choose 1,4-diiodobutane over 1,2-diiodoethane?

Comment 6: Page 5, line 181  Reference to Figure 3(c,d,e) is missing.

Comment 7: Regarding Figure 5d, the authors claim that the dielectric permittivity, mechanical properties and thermal properties of PI composite films prepared in this work are much higher than those from PI composite film using different fillers. Regarding the dielectric properties, I believe so but comparing at least the thermal properties without error bars it is not correct since the results seem quite similar. Is it possible to include the error bars in the graph?

Comment 8: Regarding the characterization methods, how did the authors prepare samples for FTIR analysis? How many scans were carried out with which resolution? Did the authors used KBr pellets or ATR accessory? Were the spectra baseline corrected and normalized? Which normalization technique was used? Regarding TGA analysis, the gas used for the analysis including the flow rate should be mentioned.

Comment 9: Regarding the references, check references 15, 26, 48 and 52 for the authors’ names and the DOI number of reference 45 is not correct.

Author Response

Response to the general comments: Thank you for your helpful suggestions. We are sorry for these grammatical errors and have carefully corrected them in the revised manuscript, hope that our revised manuscript will correctly and clearly express our intentions. We have rewritten the manuscript based on your comments and have marked the revised sections in blue color and we hope that our revised manuscript will be accepted by Molecules.

Comment 1: The introduction is well written presenting a good state-of-the-art related to the topic of the work and the references are suitable. However, in the second paragraph in page 2, the authors refer different reducing agents but not 1,2-diiodoethane from reference 47. Is there a reason as why it was not referred? What are the advantages of 1,4-diiodobutane over 1,2-diiodoethane?

Reply 1: Thank you very much for your valuable comments and helpful suggestions.

Due to our negligence, reference 47 is not mentioned in the text. We have added relevant content about 1,2-diiodoethane in the revised manuscript. Compared to 1,4-diiodobutane, 1,2-diiodoethane is expensive and the experimental conditions are more rigorous. On the contrary, 1,4-diiodobutane is liquid and can reduce GO with lower C-I bonding energy at room temperature, making the reduction more effective. In addition, 1,4-diiodiodibutane is cheap and can save experimental costs. (J. Mater. Chem. C 2015, 3. Applied Surface Science 2017, 420, 390-398).

Please refer to line 70 on page 2 of the revised manuscript.

“... In 2015, Haiquan Guo et al. used 1,2-diiodoethane to prepare graphene (IGO)/polyimide (PI) composites. The composite films had a conductivity of 2.22 S/m in this way, however the reducing agent needed to be dissolved with a polar solvent and then heated at 80℃ for 12 h for the reduction to take effect…”

Comment 2: The last two sentences of the introduction (line 80 to 83) are conclusions. It would be better to rephrase this part to address the goal of the work.

Reply 2: Thank you very much for your valuable comments and helpful suggestions. According to your suggestion, we have modified the last two sentences.

Please refer to line 83 on page 2 of the revised manuscript.

“… In this paper, reduced graphene oxide/polyimide (rGO/PI) composite films were prepared by a simple in situ polymerization method. The degree of GO reduction was successfully adjusted by varying the amount of 1,4-diiodobutane added. The high dielectric constant, low dielectric loss, high strength and high temperature resistance of the composite films were investigated by analyzing the variation of the dielectric properties, mechanical properties and heat resistance. In addition, the effect of the degree of GO reduction on the properties of the rGO/PI composite films was discussed. We believe that these high dielectric constant composite films with good mechanical and thermal properties can provide new insights into the development of capacitors, filters and the miniaturization and integration of electronic devices...”

Comment 3: Regarding the UV-Vis spectra, the shifting of the peak at 245 nm to higher wavelength is clearly evident. However, it is difficult to observe the absorbance increase of the peak as stated. How did the authors detected this occurrence since it does not seem clear from the figure?

Reply 3: Thank you for your valuable questions. We are sorry to have provided you with incorrect information. After careful examination of the experimental data, we find that there is indeed no significant change in the absorbance of the peak, so we have chosen to remove this sentence from the revised manuscript in order to improve the quality of the manuscript.

Comment 4: Page 3, line 110 – Reference to Figure 1b is missing.

Reply 4: Thank you very much for your valuable suggestion. We are very sorry for the omission of important information and the relevant content has been added in the revised manuscript.

Please refer to line 120 on page 3 of the revised manuscript:

“…However, as the proportion of reducing agent increases, the absorption peaks belonging to the π-π* transition gradually redshift to higher wavelengths, redshifted to 252 nm, 256 nm, 260 nm, and 268 nm, respectively (Figure 1b)...”

Comment 5: In page 3, why could 1,4-diiodobutane be an ideal reducing agent? Reference 47 describes the preparation of graphene (IGO)/polyimide (PI) composites using 1,2-diiodoethane as the reducing agent. How does 1,4-diiodobutane compare to 1,2-diiodoethane? Why choose 1,4-diiodobutane over 1,2-diiodoethane?

Reply 5: Thank you very much for your valuable comments. Compared to 1,4-diiodobutane, 1,2-diiodoethane is expensive and the experimental conditions are more rigorous. On the contrary, 1,4-diiodobutane is liquid and can reduce GO with lower C-I bonding energy at room temperature, making the reduction more effective. In addition, 1,4-diiodiodibutane is cheap and can save experimental costs. (J. Mater. Chem. C 2015, 3. Applied Surface Science 2017, 420, 390-398).

Comment 6: Page 5, line 181 – Reference to Figure 3(c,d,e) is missing.

Reply 6: Thank you very much for your valuable suggestion. We are very sorry for the omission of important information, and the relevant content has been added in the revised manuscript.

Please refer to line 198 on page 5 of the revised manuscript:

“… The fracture surface of pure PI was flat and smooth compared to the surrounding matrix, indicating typical characteristics of brittle fractures (Figure 3c). Compared with PI film, the fracture surfaces of composite films were relatively rough in Figure 3(d-f), which indicated that GO and PI had good interfacial interaction and GO was well dispersed in the matrix...”

Comment 7: Regarding Figure 5d, the authors claim that the dielectric permittivity, mechanical properties and thermal properties of PI composite films prepared in this work are much higher than those from PI composite film using different fillers. Regarding the dielectric properties, I believe so but comparing at least the thermal properties without error bars it is not correct since the results seem quite similar. Is it possible to include the error bars in the graph?

Reply 7: Thank you very much for your valuable and helpful questions. We have added the error bars in the graph in the revised manuscript.

Please refer to Figure 5 on page 8 of the revised manuscript:

Figure 5. (a) TGA curves, (b) Variation in dynamic mechanical properties loss tangent with temperature, (c) The T5% and Tg (d) Typical stress-strain curves, (e) The tensile strength and the Young's modulus for pure PI, GO/PI and rGO/PI composite films, (f) The dielectric permittivity, mechanical properties and thermal properties of various fillers for PI composite films.

Comment 8: Regarding the characterization methods, how did the authors prepare samples for FTIR analysis? How many scans were carried out with which resolution? Did the authors used KBr pellets or ATR accessory? Were the spectra baseline corrected and normalized? Which normalization technique was used? Regarding TGA analysis, the gas used for the analysis including the flow rate should be mentioned.

Reply 8: Thank you very much for your valuable and helpful questions. We have added experimental conditions for FTIR and TGA analyses in the revised manuscript.

Please refer to line 317 on page 9 of the revised manuscript:

“...The samples were tested in infrared transmission mode, 4000–400 cm−1. A small amount of the sample and a sufficient amount of spectral grade potassium bromide powder were taken, ground thoroughly in an agate mortar and then slowly poured into an infrared compression mould and pressed to obtain a uniform and transparent sheet. Thirty-two scans were performed at a resolution of 4 cm−1 and the spectra obtained were further Rubberband baseline corrected and no normalization was used...Thermal weight loss was measured using a TGA (Perkin Elmer Pyris1), N2 atmosphere, flow rate 30 ml/min, 40–800℃, 10℃/min.”

Comment 9: Regarding the references, check references 15, 26, 48 and 52 for the authors’ names and the DOI number of reference 45 is not correct.

Reply 9: Thank you very much for your valuable comments and helpful suggestions. We have corrected the format of the references.

Reviewer 3 Report

I would like to see formulas that can approximate the obtained dependencies

Author Response

Reviewer #3: I would like to see formulas that can approximate the obtained dependencies.

Reply 1: Thank you for your valuable advice. We are sorry for not providing this important information. We have supplemented the formulae for calculating the dielectric constant of composite films and have included a note in the revised manuscript.

Please refer to line 338 on page 10 of the revised manuscript:

“…A simple flat plate capacitor is formed using two metal plates that are parallel to each other and separated by a dielectric. A sample is placed between the two plates to measure the capacitance. The relative permittivity of the sample is calculated by the equation 1:

(1)

where C is the measured capacitance, is the relative dielectric constant of the sample, S is the area of the two parallel electrode plates, d is the thickness of the sample and k is the physical constant with a value of 9.0 * 109 N * m2/C2...”
